# The Oral–Gut Microbiota Axis as a Mediator of Frailty and Sarcopenia

**DOI:** 10.3390/nu17152408

**Published:** 2025-07-23

**Authors:** Domenico Azzolino, Margherita Carnevale-Schianca, Lucrezia Bottalico, Marica Colella, Alessia Felicetti, Simone Perna, Leonardo Terranova, Franklin Garcia-Godoy, Mariangela Rondanelli, Pier Carmine Passarelli, Tiziano Lucchi

**Affiliations:** 1Geriatric Unit, Medical Department, Fondazione IRCCS Ca’ Granda Ospedale Maggiore Policlinico di Milano, 20122 Milan, Italy; tiziano.lucchi@policlinico.mi.it; 2Respiratory Unit and Cystic Fibrosis Adult Center, Fondazione IRCCS Ca’ Granda Ospedale Maggiore Policlinico di Milano, 20122 Milan, Italy; margherita.carnevale@policlinico.mi.it (M.C.-S.); leonardo.terranova@policlinico.mi.it (L.T.); 3Interdisciplinary Department of Medicine, Section of Microbiology and Virology, School of Medicine, University Hospital of Bari, 70124 Bari, Italy; bottalico.lu@gmail.com (L.B.); marycolella98@gmail.com (M.C.); 4Doctoral School, eCampus University, 22060 Novedrate, Italy; 5Department of Medical and Surgical Sciences, Magna Graecia University, 88100 Catanzaro, Italy; felicettialessia21@gmail.com; 6Department of Food, Environmental and Nutritional Sciences, University of Milan, Via Celoria, 2, 20133 Milan, Italy; 7Bioscience Research Center, College of Dentistry, University of Tennessee Health Science Center, Memphis, TN 38163, USA; fgarciagodoy@gmail.com; 8The Forsyth Institute, Cambridge, MA 02142, USA; 9Department of Surgery, Herbert Wertheim College of Medicine, Florida International University, Miami, FL 33199, USA; 10Department of Public Health, Experimental and Forensic Medicine, University of Pavia, 27100 Pavia, Italy; mariangela.rondanelli@unipv.it; 11Endocrinology and Nutrition Unit, Azienda di Servizi Alla Persona ‘’Istituto Santa Margherita’’, University of Pavia, 27100 Pavia, Italy; 12Department of Head and Neck and Sensory Organs, Division of Oral Surgery and Implantology, Fondazione Policlinico Universitario A. Gemelli IRCCS, University Cattolica del Sacro Cuore, 00168 Rome, Italy

**Keywords:** nutrition, diet, oral microbiota, gut microbioma, aging, frailty, inflammation, mitochondrial dysfunction

## Abstract

Traditionally studied in isolation, the oral and gut microbiota are now being recognized as interconnected through anatomical and physiological pathways forming a dynamic “oral–gut microbiota axis”. Both oral and gut microbiota undergo changes with aging, characterized by a decline in microbial diversity and a shift toward potentially harmful species. The aim of this review is, therefore, to provide an overview of oral–gut communications in mediating frailty and sarcopenia. PubMed, EMBASE and Scopus databases were searched for relevant articles. We limited our search to manuscripts published in the English language. Interactions between oral and gut microbiota occur mainly through three pathways namely the enteral, the bloodstream and the fecal-oral routes. Alterations in the oral–gut microbiota axis contribute to chronic low-grade inflammation (i.e., “inflamm-ageing”) and mitochondrial dysfunction, key mechanisms underlying frailty and sarcopenia. Microbial metabolites, such as short-chain fatty acids and modified bile acids, appear to play an emerging role in influencing microbial homeostasis and muscle metabolism. Furthermore, poor oral health associated with microbial dysbiosis may contribute to altered eating patterns that negatively impact gut microbiota eubiosis, further exacerbating muscle decline and the degree of frailty. Strategies aimed at modulating the microbiota, such as healthy dietary patterns with reduced consumption of ultra-processed foods, refined carbohydrates and alcohol, ensuring an adequate protein intake combined with physical exercise, as well as supplementation with prebiotics, probiotics, and omega-3 polyunsaturated fatty acids, are increasingly recognized as promising interventions to improve both oral and gut microbiota health, with beneficial effects on frailty and sarcopenia. A better understanding of the oral–gut microbiota axis offers promising insights into nutritional interventions and therapeutic strategies for the age-related muscle decline, frailty and systemic health maintenance.

## 1. Introduction

Over the last decade, there has been an increasing research interest on the potential role of gut microbiota in influencing a wide range of conditions, including the age-related muscle decline (i.e., sarcopenia), poor nutritional status and frailty [1,2,3,4]. In particular, frailty is defined as “a clinical state in which there is an increase in an individual’s vulnerability for developing increased dependency and/or mortality when exposed to a stressor.” [5], while sarcopenia is “a progressive and generalized skeletal muscle disorder that involves the accelerated loss of muscle mass and function” [6]. Frailty and sarcopenia, despite being two distinct conditions, frequently overlap and share some physiopathological mechanisms, including those related to inflammation, oxidative stress, mitochondrial dysfunction, and hormonal dysregulation [6]. The prevalence of sarcopenia varies widely between studies and depends on the operational definition used. A recent systematic review of meta-analyses estimated a worldwide prevalence of sarcopenia of about 10–16% in older people [7]. Since 2016, according to the International Classification of Diseases, Tenth Revision (ICD-10), sarcopenia has been recognized as a disease, and it is identified by the code M62.84 [8], which is adopted by over 100 countries, including Italy, that utilize the ICD-10 system. However, in some Italian regions including the Lombardy region, the ICD-9 is still widely used in clinical practice. Indeed, it has been suggested that sarcopenia could be assigned the code 728.2 (muscle wasting and atrophy, not elsewhere classified, unspecified site) according to the ICD-9 system [9]. Unfortunately, there is no pharmacological treatment currently approved for the treatment of both frailty and sarcopenia. Therefore, strategies aimed at counteracting frailty and sarcopenia are mainly based on lifestyle interventions incorporating nutrition and exercise [7,10]. However, several drug candidates are under investigation, despite few trials recruit frail or sarcopenic subjects [11,12]. Among the various drug options under investigation, testosterone has the most accumulated evidence regarding its anabolic effects and safety profile [13]. Additionally, a mas receptor agonist, BIO101 (20-hydroxyecdysone), has emerged as a recent promising pharmaceutical with a good safety profile [13]. Just recently, the results of the phase II study (SARA-INT) indicated a significant improvement in 400-m walk test, especially for slow walkers and obese subjects after 6 to 9 months of treatment with BIO101 [14].

New therapeutic options, including apelin and irisin, are emerging and are under clinical and preclinical investigation. Apelin, upon binding to its receptor, can activate adenosine monophosphate-activated protein kinase (AMPK)-dependent pathways that promote mitochondriogenesis in the skeletal muscles of mice [15,16]. In addition to AMPK activation, apelin also influences the protein kinase B (Akt) pathway, which regulates metabolism and protein turnover in skeletal muscle during aging through multiple signaling mechanisms [11].

Irisin, on the other hand, primarily affects skeletal muscle through metabolic pathways. Exposure of C2C12 muscle cells to irisin induces overexpression of mitochondrial-specific transcription factors, such as peroxisome proliferator-activated receptor gamma coactivator 1-alpha and mitochondrial transcription factor A, enhancing mitochondrial function [17]. Both Apelin and irisin also impact satellite cells, which are essential for muscle regeneration [11]. Specifically, the apelin receptor is expressed on satellite cells, and its activation by apelin in aged mice increases satellite cell proliferation, resulting in improved muscle regeneration following cardiotoxin-induced injury [18]. Similarly, irisin promotes skeletal muscle hypertrophy by activating satellite cells and enhancing protein synthesis [11].

Given the multifactorial nature of sarcopenia, therapeutics targeting multiple pathways simultaneously are considered more likely to be effective in improving both muscle mass and function. Therefore, future prospects include therapies with multiple targets of action or combinations of multiple drugs, including combinations of pharmacological and non-pharmacological interventions [11,13]. Targeting microbiota is also emerging as a new therapeutic option for sarcopenia. Interestingly, *E. Faecalis* has been recently identified as a potential new inducer of sarcopenia both in animal models and in humans [19]. Consequently, bacterial quorum sensing peptides (e.g., iAM373), which are produced by *E. Faecalis* opened new therapeutic perspectives in the field of sarcopenia [19].

The oral microbiota has traditionally received less attention compared to the gut microbiota. However, in recent years, there has been a growing research interest on the oral microbiome [20,21], making it one of the five research priorities of the Human Microbiome Project [22]. Both the oral cavity and the gut host some of the most dense and diverse microbial communities in the human body [23]. Although they are physiologically distinct environments, they are directly connected and can influence each other in a variety of ways [23,24]. The gut microbiota has been historically regarded as the central core of microbial health due to its critical role in digestion, nutrient absorption, and immune function [25]. The oral microbiota also plays a major role in shaping both oral and systemic health, as it serves as the primary gateway for nutrients and pathogens entering the digestive system [20,26]. Studies are now beginning to reveal how factors such as diet, oral hygiene, and the environment can influence the composition and function of the oral microbiota, and how these shifts may be linked to both oral and systemic diseases [27,28]. In this context, both oral and gut microbiota alterations have been associated with the so-called inflammaging process, characterized by a state of chronic, low-grade inflammation [29] and considered a hallmark of ageing [30]. In turn, inflammaging represents the substrate for the development of malnutrition, frailty and sarcopenia [6,31,32]. Furthermore, poor oral health and associated dysbiosis may further increase the risk for malnutrition, frailty and sarcopenia through several pathways including decreased chewing ability, bolus formation and infections [6,33,34,35]. The aim of this review is, therefore, to provide an overview of oral–gut communications in mediating frailty and sarcopenia.

## 2. Overview on the Oral and Gut Microbiota Changes with Aging

The oral microbiota is a complex and dynamic ecosystem comprising a wide array of microorganisms, including bacteria, viruses, fungi, and archaea [36]. One of the most comprehensive resources for studying the human oral microbiome is the Human Oral Microbiome Database (eHOMD) (Figure 1) [37].

Bacteria are the most abundant microorganisms in the oral cavity, with over 700 identified species [36]. The oral environment includes both oxygen-rich (e.g., supragingival surfaces) and oxygen-poor areas (e.g., subgingival surfaces), allowing for the coexistence of both aerobic and anaerobic bacterial species [38]. The primary bacterial phyla present include Bacillota, Actinomycetota, Pseudomonadota, Bacteroidota, and Spirochaetota, with *Streptococcus* being the dominant genus [39]. Within the Bacillota phylum, *Streptococcus* species are among the most prevalent and are closely associated with dental plaque and caries [40]. *Lactobacillaceae*, another Bacillota member, is also implicated in tooth decay [41]. Actinomycetota, such as *Actinomyces* and *Bifidobacterium*, contribute to the oral microbiota balance [42]. Pseudomonadota like *Neisseria* are commonly found in the saliva and oral mucosa [43], while Bacteroidota such as *Prevotella* species are linked to gum and periodontal diseases [44]. Spirochaetota, particularly *Treponema*, are associated with advanced periodontal conditions [45]. Fungi, mainly from the *Candida* genus (especially *C. albicans*), are present in small numbers but can become pathogenic under certain conditions, such as immunosuppression, leading to oral candidiasis [46].

The oral cavity also harbors various viruses, including phages and the human papilloma virus (HPV), the latter being associated with oral cancer [47,48]. Although less abundant, Archaea such as *Methanobrevibacter* are also part of the oral microbiota [49,50], and some studies link them to dental plaque, periodontal disease, and halitosis due to their methane production [50,51,52].

### 2.1. Oral Microbiota Changes with Aging

Once established, the oral microbiota tends to remain relatively stable [53], but several factors including diet, oral hygiene, and hormonal changes can lead to oral dysbiosis across life [27]. With aging, microbial diversity typically declines [54]. This shift often involves a reduction in beneficial bacteria and an increase in potentially harmful species, contributing to conditions like gum disease, dry mouth, and tooth decay [54]. For example, *Neisseria* levels tend to decrease after age 40, while levels of *Lactobacillaceae*, *Streptococcus anginosus*, and *Gemella sanguinis* increase after age 60 [55]. The use of dentures or other dental appliances can also significantly alters the microbial community during old age [54]. Differences in oral microbiota composition and diversity have been observed between denture wearers and individuals with natural teeth. In denture users, *Bacillota* and Actinomycetota are often predominant on denture surfaces and oral mucosa [56]. The presence of residual natural teeth also affects microbiota composition in denture wearers [56]. Older adults with fewer natural teeth tend to have oral communities dominated by species like *Prevotella histicola*, *Veillonella atypica*, *Streptococcus salivarius*, and *Streptococcus parasanguinis* [54,57]. Interestingly, “toothy” centenarians—those retaining 20 or more natural teeth—exhibit a more diverse oral microbiota [58], rich in phyla such as Spirochaetota and Synergistota, and includes genera like *Aggregatibacter*, *Prevotella*, *Campylobacter*, *Anaeroglobus*, *Selenomonas*, *Fusobacterium*, and species such as *Porphyromonas endodontalis* [58]. Furthermore, in their dental plaque, species like *Bifidobacterium*, *Scardovia*, *P. gingivalis*, *Tannerella forsythia*, and *Prevotella intermedia* are particularly abundant [58]. On the other hand, the oral microbiota composition of edentulous centenarians shows an enrichment of Bacillota and Actinomycetota at the phylum level and of *Streptococcus* spp at genus level, found in both dental plaque and saliva [58]. Table 1 shows the main changes in oral and gut microbiota with aging.

### 2.2. Gut Microbiota Changes with Aging

The gut microbiota undergoes significant development from birth to age three, after which it stabilizes [59,60]. The adult human gut microbiota is largely composed of Bacteroidota and Bacillota, which together make up more than 90% of phylogenetic types [58]. These are further divided into over 100 bacterial species [61]. However, aging prompts changes again, particularly in species diversity (Table 1) [62,63,64,65]. Older adults seem to experience a decreased Bacillota/Bacteroidota ratio, a reduction in *Bifidobacterium* populations, and an increase in certain Pseudomonadota [66] (Figure 2).

Interestingly, centenarians tend to have more diverse gut microbiota compared to the average older population. While Bacillota remain prominent, the Bacillota/Bacteroidota ratio is generally lower. In Chinese centenarians, higher levels of beneficial *Lactobacillaceae*, known for their anti-inflammatory and antioxidant properties, have been reported [69]. In Sardinian centenarians, *Faecalibacterium prausnitzii* and *Agathobacter rectalis* (formerly *Eubacterium rectale*) are depleted, while *Methanobrevibacter smithii* and *Bifidobacterium adolescentis* are enriched compared to younger individuals [70], while only mild increases in Pseudomonadota, particularly *Escherichia coli* have been documented [67,70,71]. Age-related microbial shifts are therefore influenced not only by chronological age but also by several factors such as the use of multiple medications including antibiotics, proton pump inhibitors, and laxatives, chronic conditions, immune system aging (i.e., immunosenescence), increased gut permeability to lipopolysaccharides, and changes in diet and lifestyle, but also by individual genetic predisposition [62,63,72,73,74,75].

## 3. Oral and Gut Microbiota Interactions

The connection between oral and gut microbiota, often referred to as the “oral–gut microbiota axis”, is an emerging area of research gaining significant attention [76]. Traditionally studied in isolation, the oral and gut microbiota are now understood to be interconnected, influencing each other through various anatomical and physiological pathways [77].

Since the mouth and the gut are continuous parts of the gastrointestinal tract, oral microbes can migrate and colonize the gut [77]. This interaction is also part of broader microbial communication networks, including the oral–lung–gut axis, where oral bacteria may reach the lung via inhalation and later influence gut microbiota through immune and microbial pathways [78,79]. Costa et al. [80] recently identified 61 shared amplicon sequence variants (ASVs) in both oral and gut microbiota in 96% of the participants examined. Notably, 26 of these ASVs (from 18 genera) were found across both children and adults, indicating a persistent colonization. However, as above-mentioned, microbial composition shifts with age, with significant changes after age 45. Importantly, 62% of shared ASVs were more abundant in the oral cavity, suggesting a predominant oral-to-gut microbial transfer [80].

Research indicates that the oral and gut microbiota communicate with one another through a complex system known as the oral-gut axis. Oral-to-gut and gut-to-oral microbial transmissions and remodeling of both habitats can eventually modulate the pathogenesis of several diseases [77]. This two-way bidirectional interaction takes place through several pathways, including the bloodstream, saliva, and fecal–oral routes [77].

### 3.1. The Enteral Route

About 1–1.5 L of saliva are swallowed along with food each day, providing a vehicle for oral bacteria to enter the digestive tract [23]. Physiologically, many microbes are unable to survive harsh environments characterized by the presence of gastric and bile acids. However, certain life stages (e.g., infancy and aging) and some conditions like gastrointestinal disease, proton pump inhibitors and antibiotic use, can weaken gastrointestinal barriers, allowing oral bacteria to reach and colonize the gut. [23,77,81]. Furthermore, biofilm formation can provide protection to oral bacteria such as *Streptococcus mutans*, helping them survive harsh environments [24,82]. Several oral pathogens including *Porphyromonas gingivalis*, *Klebsiella* spp., *Helicobacter pylori*, *Streptococcus* spp., *Veillonella* spp., *Parvimonas micra*, and *Fusobacterium nucleatum* can survive acidic environments and reach the intestine [24,83]. Even under normal conditions, bacteria like *Prevotella* have been detected in both saliva and fecal samples [84]. Additionally, *Helicobacter pylori* infection can disrupt the oral and gastric microbial balance, promoting overgrowth of oral bacteria in the gut like *Fusobacterium nucleatum* and *Porphyromonas gingivalis* [24,85].

Bile acids also play a crucial role. While they aid digestion by emulsifying fats, altered bile acid flow (e.g., in gastroesophageal reflux disease [GERD]) can affect the microbiota of the mouth, esophagus, and gut [86,87]. In reflux disorders, bile acids and gastric acid together damage the esophageal lining. Bile acids alterations have also been associated with inflammation, dysbiosis, and even gastrointestinal cancers [88,89,90,91]. Bile acids can be protonated at acidic pH showing synergistic detrimental effects with gastric acid [92,93]. This acidic environment fosters the growth of acid-tolerant oral bacteria, reduces beneficial species with consequent dysbiosis, and contributes to dental erosion, caries, and periodontitis [94,95,96,97,98] as well as inflammation and increased susceptibility to opportunistic infections [88,89]. Notably, increased bile acids, including potentially carcinogenic ones like glycocholic acid, have been found in the saliva of patients with GERD [87,99,100]. The relationship is bidirectional: oral dysbiosis may promote or be promoted by altered bile acids [101] and gastric acid reflux [102]. Studies also suggested associations between poor oral health (e.g., tooth loss and poor oral hygiene) and gallstone disease, and oral pathogens like *Pyramidobacter* have been found in bile samples of gallstone patients, suggesting the oral microbiota may impact gallbladder health [103,104,105].

### 3.2. The Bloodstream Route

Oral microbes can enter the bloodstream through mechanical actions like brushing, chewing, or dental procedures, particularly when gum tissues are inflamed or damaged [23,24]. This allows bacteria to travel systemically, including to the gastrointestinal tract. Once in the bloodstream, oral pathogens can contribute to systemic inflammation and intestinal dysbiosis. *P. gingivalis*, *F. nucleatum*, and *Streptococcus* species have all been implicated in this process [23,24]. Animal studies show that *F. nucleatum*-induced periodontitis can cause gut inflammation [106], while in humans, *F. nucleatum* has been found in colon tumors, associated with chemoresistance and poor outcomes [107]. These bacteria can also damage the intestinal epithelial barrier, allowing microbial translocation and metabolic leakage [24]. For instance, *P. gingivalis* produces gingipains, enzymes that degrade tight junctions in the colon, compromising barrier integrity [23] and promoting systemic inflammation [108,109,110]. Some oral bacteria, like *Streptococcus salivarius*, may also play a dual role by regulating inflammation while maintaining mucosal health via modulation of immune factors like the transcription factor nuclear factor kappa B (NF-κB) [20].

Antibiotic use, frequent in dentistry, also plays a significant role [111]. It can disrupt the gut microbiota, increase antimicrobial resistance, and worsen systemic health outcomes, especially in older people [112]. Overuse of antibiotics can, in fact, negatively affect T helper 17 (Th17)/regulatory T (Treg) cells balance, and increase the abundance of periodontitis-associated pathogens by reducing probiotics and increasing levels of pro-inflammatory cytokines, finally exacerbating periodontitis itself [24]. Probiotics are often used to counteract antibiotic-induced dysbiosis. However, evidence shows limited effectiveness, especially in restoring microbiota diversity post-antibiotic treatment [113,114].

While less studied, associations in the reverse direction (i.e., gut-to-oral) have also been documented [23]. For instance, a meta-analysis indicated IBD was associated with a significantly higher risk of periodontitis compared to non-IBD patients [115]. A study conducted in mice models of Crohn’s disease showed that periodontitis naturally develops in these models and strongly correlates with the severity of ileitis [116]. Additionally, systemic inflammatory conditions may impair the immune barrier functioning of the oral mucosa, resulting in heightened inflammation and greater vulnerability to periodontal disease [23,117,118]. In type 2 diabetes, hyperglycemia-driven changes in the periodontal microbiota and amplified inflammatory responses in the mouth can increase susceptibility to periodontitis [117,119,120]. In a similar way, type 1 diabetes affects oral physiology by elevating both local and systemic inflammatory mediators and through chronic hyperglycemia, promoting the formation of advanced glycation end-products, compounds formed by non-enzymatic glycation and oxidation of proteins, lipids, and nucleic acids [121]. Thus, these findings, although being preliminary, suggest a bidirectional relationship between periodontal disease, oral dysbiosis, and systemic diseases.

Translocation of microorganisms from gut to the oral cavity can also take place through fecal-oral transmission, either through direct contact or indirectly via contaminated food and beverages. Hands play a crucial role as carriers, facilitating the transfer of fecal and oral microorganisms both within households and between individuals [24,122].

### 3.3. Fecal-Oral Route

Microbes can also travel from the gut to the oral cavity via fecal–oral transmission, especially through contaminated hands, food, or water. This is particularly common in settings with poor sanitation [77,122]. Hands serve as carriers of microbes from the gut and mouth and facilitate transmission between individuals [122]. This route poses higher risks for immunocompromised individuals and patients undergoing radiation therapy [77], which can worsen oral dysbiosis and promote colonization by pathogens such as *Candida* and *Enterobacteriaceae* [123,124]. Fecal–oral transmission is a major pathway for enteric viruses like *Hepatitis A* and *E*, which can disrupt the gut microbiota [125,126,127,128] and have also been linked to *H. pylori* infections [129].

## 4. Influence of the Oral–Gut Axis on Frailty and Sarcopenia

Frailty and sarcopenia share some common physiopathological mechanisms mainly related to the physical function domain [6,130]. Among all, inflammation and mitochondrial dysfunction, considered hallmarks of ageing [30], are among the most promising and relevant mechanisms underlying the two conditions [32,35,131]. Gut microbiota alterations have long been recognized at the basis of frailty and sarcopenia [2], while oral microbiota dysbiosis has been less explored, also in this case. However, as discussed above, recent evidence started to point out the role of the oral–gut communication axis, with alterations in these microbiota resulting in the common burden of inflammation, mitochondrial dysfunction and consequent oxidative stress. The translocation from the oral cavity to the gut of of *P. gingivalis*, including its lipopolysaccharide (LPS), can promote gut dysbiosis and inflammation by upregulating toll-like receptor 2 (TLR2), tumor necrosis factor alpha (TNF-α), and interleukin-17 (IL-17), also contributing to hepatic inflammation and liver fibrosis in metabolic dysfunction-associated fatty liver disease [24]. Even slight increases in microbiota-derived LPS in the circulation have been reported as important drivers of low-grade inflammation [24]. When *P. gingivalis* colonizes the gut, it enhances gut permeability and significantly increases endotoxemia:(1)By enhancing LPS in the bloodstream, which subsequently induces the upregulation of flavin-containing dimethylaniline monooxygenase 3 expression (FMO3) and elevates circulating trimethylamine *N*-oxide (TMAO) concentrations, resulting in metabolic dysregulation, gut dysbiosis and inflammation [109,110];(2)By downregulating the expression of tight junction protein cytosolic zonula occludens 1 (ZO-1) and occludin in the small intestine, thereby increasing intestinal permeability [109].

Additionally, *P. gingivalis* seems to promote IL-6 expression through the janus kinase 2/glycogen synthase kinase 3 beta/signal transducer and activator of transcription 3 (JAK2/GSK3-β/STAT3) pathway, which is associated with carcinogenesis and oral squamous cell carcinoma [132]. *P. gingivalis* has also been reported as inhibiting mitochondria-mediated cell apoptosis [132]. It disrupts the JAK1/Akt/STAT3 signaling pathway, leading to suppression of the pro-apoptotic protein BAD on the mitochondrial membrane and an increased B-cell lymphoma(Bcl)-2/Bcl-2-associated X protein ratio. This alteration results in impaired mitochondrial membrane permeability and decreased cytochrome c release, thereby inhibiting the activation of downstream apoptotic effectors caspase-9 and caspase-3. Additionally, *P. gingivalis* upregulates microRNA-203 expression, which suppresses suppressor of cytokine signaling 3, a key regulator within the JAK1/Akt/STAT3 pathway, further promoting apoptosis inhibition. It also secretes nucleoside diphosphate kinase, which inhibits apoptosis mediated through the purinergic receptor P2X7 [132].

Furthermore, recent studies highlighted the prominent role of metabolites like the short-chain fatty acids (SCFAs) acetate, propionate and butyrate produced by the gut microbiota from undigested dietary fiber [133]. These SCFAs may be translocated from the gut to the oral cavity through the bloodstream, influencing oral pH [133,134]. The oral microbiota may, in turn, transmit signals back to the gut, thereby modulating gastrointestinal health. This implies a potential bidirectional interaction between oral and gut microbiota, mediated by microbial metabolites, which may contribute to the regulation or disruption of microbial homeostasis [133,135]. The oral microbiota generates SCFAs through carbohydrate or amino acid metabolism too [133], although their concentration is generally lower than in the gut [136,137]. SCFAs have recognized anti-inflammatory properties, including the ability to reduce the production of reactive oxygen species (ROS) and myeloperoxidase by neutrophils and promote their apoptosis [133]. The anti-inflammatory effects of SCFAs also include the suppression TNF-α and IL-12, the enhancement of mucosal barrier integrity, as well as the restoration of the Treg/Th17 balance [24,132,138]. Some amino acids metabolized in the gut, such as arginine, can generate compounds like nitric oxide (NO), which may exert antibacterial effects in the oral cavity [139]. However, the role of SCFAs in the oral cavity is more nuanced. While they are part of normal metabolism, excessive SCFAs production at oral level can be a marker of oral dysbiosis, triggering soft tissue damage and heightened inflammation [133,140]. Unlike the oral mucosa, the gut mucosa appears more resilient, tolerating higher SCFAs concentrations without cellular damage, probably because of a better adaptive capacity [133]. Beyond their anti-inflammatory properties, SCFAs have some potential effects in muscle metabolism [141,142,143]. In fact, acetate is mostly utilized by muscle cells to produce energy [144] and SCFAs production has been correlated with muscle anabolism while their depletion may promote anabolic resistance [145,146,147].

Chronic, low-grade inflammation, is a critical mediator linking oral pathogens to systemic diseases [148] and it is a well-known contributing factor in the etiopathogenesis of malnutrition, frailty and sarcopenia [6,31,32,149]. Pathogens originating from the oral cavity can elicit immune responses with the consequent release of pro-inflammatory cytokines, including TNF-α, IL-1β, and IL-6, especially in the older population with oral diseases like periodontitis [150]. Increased levels of oral pathogens like *Streptococcus mutans*, *Porphyromonas gingivalis*, *Campylobacter concisus*, and *Fusobacterium nucleatum* in the gut have been observed in conditions like inflammatory bowel disease (IBD), human immunodeficiency virus, liver cirrhosis, and colorectal cancer [23,24,108,151,152,153,154], conditions frequently characterized by a high prevalence of frailty and sarcopenia [6,155,156,157,158]. Some evidence supporting oral–gut communication, including mechanisms that may influence frailty and sarcopenia, comes from studies on IBD. In IBD, bile acid malabsorption and receptor alterations may facilitate the migration of oral pathobionts to the gut [159]. Proposed mechanisms by which oral bacteria contribute to gut dysbiosis and IBD include [24,108]:Disruption of intestinal barriers. *P. gingivalis* and *K. pneumoniae* and consequently gut inflammation, have been indicated as downregulating the expressions of tight junction protein 1 and occludin. Additionally, the secretion of gingipain proteases disrupts the mucus layer’s function and integrity by degrading intestinal mucus and inhibiting mucus shedding locally, as well as breaking down junction-associated proteins like the cytosolic ZO-1 [23].LPS-triggered inflammation. *F. nucleatum*, *K. pneumoniae*, and *P. gingivalis* can trigger the release of LPS [108,132]. LPS from *P. gingivalis* activates the NF-κB pathway and Caspase-1 inflammasome, resulting in increased IL-1β and IL-18 production [132], which drive intestinal inflammation and can cross the blood–brain barrier to promote neuroinflammation by activating microglia [138].T cell imbalances. *F. nucleatum* and *Candida albicans* can disrupt the balance between Th1/Th17 cells, further inducing inflammatory reactions [108]. *P. gingivalis* and *F. nucleatum* can trigger overproduction of pro-inflammatory cytokines such as IL-6, IL-8, IL-1β, TNF-α, IL-17, CXC motif chemokine ligand 10 (CXCL10), and IL-23 via TLR2, TLR4, Th17 cells and myeloid differentiation primary response 88 (MYD88) signaling [23,138,160,161]. In turn, the abnormal release of several of these pro-inflammatory cytokines and chemokines, including IL-6, TNF-α, and CXCL10, has been independently associated with frailty [162] and sarcopenia [163].Inflammasome activation and immune dysregulation. Pathogenic microorganisms can also influence the oral–gut microbiota axis through immune pathways [24]. Imbalances in the oral microbiota can influence gut-associated immune cells, triggering immune responses negatively impacting both oral and gut health [24]. Oral pathogens, like *Klebsiella* and *Enterobacter*, when colonizing the gut, can activate the inflammasome and induce inflammation in colonic mononuclear phagocytes, disrupting the intestinal immune environment [24,164]. *Klebsiella* species also show adaptive capacity to distant mucosal sites such as the gut through sophisticated virulence strategies [165]. *Streptococcus gordonii* has been found to hinder macrophage-mediated destruction of *Candida albicans*, further contributing to immune system dysregulation [166]. Beyond reducing Th17 cell levels, oral dysbiosis can also reduce fecal immunoglobulin A (IgA), altering the M1/M2 macrophage balance, further promoting chronic inflammation. Oral microbiota dysbiosis can also be responsible for metabolic alterations by increasing lactate levels and reducing beneficial metabolites like succinate and n-butyrate, exacerbating gut dysbiosis [167]. The presence of oral bacteria in the gut can lead to mucosal and intestinal epithelial barrier damage by influencing lamina propria macrophages and increasing IL-1β levels through the overstimulation of the inflammasome [168]. This is particularly evident in periodontal disease, in which salivary-induced dysbiosis alters gut microbiota and exacerbates colitis with the consequent damage of the mucosal barrier [169]. Notably, it has been reported that about 30% of individuals with IBD show oral symptoms that may precede gastrointestinal manifestations, indicating a bidirectional relationship where systemic inflammation in IBD can alter oral microbiota and intensify oral inflammation [170].

Dental interventions aimed at restoring oral function in older adults may also inadvertently lead to bacteremia, potentially exacerbating systemic inflammation [171]. Microbiota alterations observed with aging, have been identified as triggers of inflammation and mitochondrial dysfunction [1]. For instance, mitochondria respond to the microbiota and its metabolites such as SCFAs and secondary bile acids, by modulating energy production, redox homeostasis, and immune responses. [1,142]. In turn, inflammation also impacts mitochondrial function, as damage-associated molecular patterns (DAMPs) released from dysfunctional mitochondria trigger further cytokine/chemokines, NO and ROS production, perpetuating a cycle of cellular stress and systemic inflammation, finally resulting in muscle wasting [1,131] (Figure 3).

DAMPs, damage-associated molecular patterns; ETC, electron transport chain; NO, nitric oxide; PAMPs, pathogen-associated molecular patterns; ROS, reactive oxygen species.Recently, some studies have started to investigate the associations between salivary microbiota and frailty [172]. Ogawa et al. [173] compared the salivary microbiota in nursing home (NH) residents, usually characterized by an increased degree of frailty, and community-dwelling older people. In particular, despite several limitations of their study, Ogawa et al. [173] found a higher relative abundance of *Actinomyces*, *Streptococcus*, *Bacilli*, *Selenomonas*, *Veillonella*, and *Haemophilus* taxa, and a decreased relative abundance of *Prevotella*, *Leptotrichia*, *Campylobacter*, and *Fusobacterium* in NH residents. Another study from Wells et al. [174], reported a positive association between frailty and decreased salivary microbiota diversity in a UK cohort of adult twins.

On the other hand, the exploration of the role of oral microbiota on muscle mass and function is still in its infancy, with very few published studies.

Research is advancing in this direction, with an ongoing study namely the Saliva and Muscle (SaMu) study aimed at addressing this knowledge gap by using a salivary multi-omics approach to clarify associations between the oral microbiome and sarcopenia in older people [172]. Beyond studying microbial composition, investigators of the SaMu study suggest that investigating saliva metabolomics, proteomics or peptidomics concerning sarcopenia may provide deeper insights into the oral microbiome and help identify potential mediators involved in the oral-muscle interactions [172]. In fact, in line with the mechanisms illustrated above, some metabolites measured in the saliva like NO and inflammatory cytokines (e.g., IL-6) have been indirectly associated with frailty and/or sarcopenia [175,176]. Yuzefpolskaya et al. [177], recently explored associations of the sarcopenia index, a surrogate biomarker for skeletal muscle mass, with inflammation, gut and oral microbiota in patients with heart failure, left ventricular assist device, and heart transplant. They found the natural logarithm of the sarcopenia index was inversely correlated with inflammation, and positively correlated with both gut and oral microbial diversity, evaluated through the Shannon index.

Some studies explored associations of oral conditions characterized by oral dysbiosis and inflammation, like dental caries and periodontal disease, with sarcopenia and its components. Yang et al. [178] investigated the association of dental caries, starting with microbial shifts within the complex biofilm [179], with muscle mass, muscle strength, and sarcopenia in a large sample of people aged 50 years from China. They also described the gut microbial composition and diversity in the context of severe dental caries and sarcopenia. The authors found positive associations of dental caries with low muscle strength and sarcopenia, while no significant association was found with low muscle mass. Furthermore, severe dental caries were positively associated with both higher alpha-diversity and beta-diversity indices, with the severe dental caries group and the sarcopenia group overlapping with 11 depleted and 13 enriched genera.

Periodontal disease, which is frequently associated with oral microbiota dysbiosis, has also been associated with sarcopenia and its components [180,181,182]. In particular, it has been suggested that periodontal disease might be implicated in the decline of physical performance (determining the severity of sarcopenia) through systemic inflammatory or immunological responses to oral microbiota [151,180,183]. In the British Regional Heart Study (BRHS) cohort study [184], periodontal pocket depth (a measure reflecting the current status of periodontal disease) has been associated with a decline in both the chair stand test and gait speed, key components of sarcopenia. Periodontal disease has also been associated with a faster decline in handgrip strength [182] and physical frailty in later life [185].

Dental caries and periodontal disease are considered the leading cause of tooth loss through microbial alterations at the oral level. Tooth loss has been in fact associated with the presence of cariogenic bacteria like Bacillota [186]. In turn, tooth loss has also been widely associated with sarcopenia, frailty and its components [178]. In the Health, Aging and Body Composition (Health ABC) study [187], total tooth loss was associated with slower gait speed, while in the BRHS study [184], tooth loss during the follow-up period was associated with a decline in the chair stand test. In both the Health ABC [187] and BRHS [184] studies, dry mouth (i.e., xerostomia) was associated with declines in physical function. Beyond its role in the bolus formation, saliva plays a pivotal role in maintaining microbial balance. Xerostomia is very common with advancing age, affecting up to 50% of older adults and it is often due to medications or systemic diseases [188]. Saliva normally helps maintain pH, microbial transportation, and contains antimicrobial proteins [24,188]. However, in conditions like xerostomia, the decreased salivary flow can lead to an increased colonization by acidogenic and pathogenic bacteria (e.g., *Streptococcus* and *Fusobacterium* species), which further exacerbates inflammatory response with the release of IL-6, IL-8, IL-17, IL, 23, IL-1β, TNF-α [188].

Because of its modulating effects on the gut microbiota, the oral microbiota has also been implicated in the development of obesity [3] through various mechanisms including metabolic and inflammatory dysregulation in adipose tissue, systemic inflammation and modifications of taste perception, food preferences and eating behaviors [189]. As a consequence, obesity may also lead to modifications in microbiota composition by altering both overall species diversity and the ratio of Bacillota/Bacteroidota [190]. In this context, it is well-established that obesity accelerates the progression of sarcopenia by intra- and inter-muscular fat infiltration and increased inflammation, characterizing the so-called “sarcopenic obesity” condition [191,192,193,194].

## 5. Dietary and Exercise Strategies Targeting the Oral and Gut Microbiota and Their Effects on Frailty and Sarcopenia

### 5.1. Dietary Strategies

Diet is a widely acknowledged pivotal factor in modulating the gut microbiota. Beyond oral health care, including both oral hygiene and dental treatments, diet also plays a significant role in the modulation of the oral microbiota. At the oral level, gum inflammation and the development of periodontal disease are promoted by poor oral hygiene, tobacco smoking, stressful conditions and depression, as well as dietary habits, malnutrition, excessive alcohol consumption, and the presence of oral pathogens [195]. Specific dietary patterns and nutrient intake can trigger or modulate immune-mediated inflammatory responses. For instance, insufficient consumption of dairy products, fruits and vegetables, dietary fiber, calcium, antioxidants, and essential fatty acids can promote pro-inflammatory processes that may contribute to the development of periodontal disease [195]. Furthermore, oral problems have been identified as risk factors for the anorexia of aging and in turn as frequent underlying causes of malnutrition, frailty and sarcopenia [6,33,34,35]. Poor oral health characterized by tooth loss because of periodontitis and other dental conditions may affect chewing ability [35,195]. In turn, diminished chewing ability negatively impacts the intake of various food groups and essential nutrients [35,195], as well as nutrient utilization, potentially contributing at (1) altering the microbiota equilibrium [2], (2) increasing inflammatory status [33,196], and (3) the risk for malnutrition, frailty, and sarcopenia [33,34]. In this context, periodontal disease has been associated with a faster decline in handgrip strength [182], and some studies reported associations between chewing difficulties and frailty [33]. It has also been suggested that the number of remaining teeth may be reflective of lifetime exposure to inflammation, as well as of musculoskeletal decline and consequent disability [182]. In fact, it is reasonable that long-lasting inflammatory status during early life, maybe not directly measurable, could have influenced one’s musculoskeletal functional reserves [182,197]. Each missing tooth can be seen as an indicator of past periodontal or dental infections and carious lesions. Accordingly, the more teeth an individual is missing, the longer he/she likely experienced oral problems earlier in life [182], which are in turn associated with a subsequent reduced muscle strength later in life [182]. Micronutrient deficiencies, even subtle, can feed oxidative stress and consequently inflammation [35]. Therefore, the lack of certain essential nutrients because of oral problems can further exacerbate nutritional status, sarcopenia and frailty [35]. Micronutrient deficiencies have also been associated with alterations in the mineralization process, increasing the risk for dental caries [198]. Additionally, poor nutritional status can increase the severity of oral infections [199].

Fermentable carbohydrates, including simple sugars and starches, act as key energy sources for bacterial metabolism [200]. Excessive carbohydrate intake, especially in refined sugars, promotes the accumulation of dental plaque, which in turn facilitates the growth of cariogenic bacteria such as *Streptococcus mutans* and *Fusobacterium nucleatum* [201,202]. Excessive alcohol intake alters oral microbiota balance, by increasing the abundance of Gram-positive species like *Streptococcus mutans*, while reducing populations of taxa such as *Fusobacterium* spp. [203,204]. On the other hand, certain nutrients, including dietary fats and vitamin C, seem to support the abundance of *Fusobacterium* spp., whereas dietary fiber and dairy products have been associated with oral microbial homeostasis [28,203,205,206,207]. Although the relationship between carbohydrate consumption and periodontal disease has been less extensively investigated [208], emerging data indicate that diets rich in whole grains, rather than in refined carbohydrates, may offer protective effects for periodontal health [209].

Changes in gut microbiota, mainly due to several clinical conditions including oral diseases, may impact the bioavailability of dietary amino acids [1,210,211]. Dietary and endogenous proteins are hydrolyzed in the gastrointestinal tract into amino acids and peptides [212,213]. These lasts are then released and support the growth and survival of bacteria in the gastrointestinal tract [214] with also regulatory functions in energy and protein homeostasis [215,216]. Protein intake appears to contribute to both gut [217,218] and oral microbiota diversity [219,220] in humans. Observational studies reported associations of poor oral health with poor dietary diversity [221,222,223] and with a low protein intake [220,224] in older adults. Inadequate protein intake may weaken tooth structure, supporting tissues and may impair wound healing, reducing individual’s ability to cope with oral pathogens [220]. Additionally, protein deficiency can lead to decreased salivary flow and changes in saliva composition, diminishing its protective functions [225]. Protein-energy malnutrition has also been associated with salivary gland atrophy and with enamel hypoplasia, increasing susceptibility to dental caries [225,226]. Cattaneo et al. [219] in a sample of 59 apparently healthy adults reported associations between protein intake and several bacterial taxa, including *Selenomonas*, *Johnsonella*, *Prevotella*, *Peptostreptococcus*, and *Actinomyces*. Fluitman et al. [220], in a 6-month randomized controlled trial, evaluated the effects of a dietary counseling aimed at increasing protein intake to ≥1.2 g/kg adjusted body weight/day on oral microbiota and oral health in community-dwelling older adults. They found moderate shifts in oral microbiota diversity, while the abundance of individual bacterial taxa was not influenced. Regarding gut microbiota, Farsijani et al. [227], in a large cross-sectional analysis of the Osteoporotic Fractures in Men Study involving older people, found an association between higher protein intakes from either animal or vegetable sources with higher gut microbiome diversity. In particular, they found greater abundance of several genus-level ASVs, including *Christensenellaceae*, *Veillonella*, *Haemophilus*, and *Klebsiella* in older participants with higher protein consumption, while *Clostridiales bacterium DTU089* and *Desulfovibrio* were more abundant in older participants with a lower protein intake. In a randomized controlled trial performed in overweight human adults, protein supplementation led to a significant change in bacterial metabolism toward amino acid degradation and fermentation [228]. Animal studies have shown that a greater production of branched-chain amino acids (BCAAs) by the gut microbiota—often linked to a balanced Bacillota/Bacteroidota ratio—is associated with enhanced insulin sensitivity and increased protein synthesis [4,229]. However, human studies gave controversial results, since elevated serum levels of BCAAs have been generally associated with insulin resistance [230]. The cross-talk between protein intake and the gut microbiota composition is, therefore, quite complex and not yet fully understood, probably influenced by factors such as protein quality and microbiota metabotype [4,231]. It has been therefore suggested that microbiota-mediated anabolic responses to protein intake/supplementation, may differ on an individual basis [4].

Most studies are focusing on the manipulation of microbiota as a possible countermeasure against sarcopenia and frailty. The main approaches, currently under investigation, include probiotics, prebiotics and symbiotic supplementation [232]. Growing interest also surrounds the use of probiotics and prebiotics to support oral microbiota health. Probiotics are defined as “live microorganisms which, when administered in adequate amounts, confer a health benefit on the host” [233]. The most studied in the aging process are those belonging to *Bifidobacteriaceae* and *Lactobacillaceae* strains [234,235], which help maintain microbial eubiosis, modulate immune responses, and produce antimicrobial agents [236,237]. They exert their effects by competing for epithelial adhesion sites, synthesizing bacteriocins, enhancing immune mechanisms such as secretory IgA production, and downregulating proinflammatory cytokines and matrix metalloproteinases (MMPs) [21,236,237]. These actions contribute to the inhibition of pathogenic bacterial growth and modulation of immune responses both locally and systemically [21,236,237].

Furthermore, probiotics may alleviate halitosis by reducing sulfur-producing bacteria on the tongue, balancing oral pH, and enhancing salivary secretion in individuals with xerostomia [27]. Despite growing evidence supporting the oral health benefits of probiotics, additional clinical research is needed to establish their long-term efficacy and optimal therapeutic protocols. Prebiotics are fermented non-digestible compounds supporting the proliferation of health-promoting bacteria [238], while synbiotics are supplements combining probiotics and prebiotics often working synergistically to enhance their efficacy [239]. While the effects of probiotics on periodontal disease have been widely investigated, the role of prebiotics remains underexplored [148,240]. Prebiotics are typically composed of carbohydrate-based compounds such as fructo-oligosaccharides and galacto-oligosaccharides, but can also include non-carbohydrate substances like polyphenols and polyunsaturated fatty acids [241]. These compounds selectively stimulate beneficial taxa such as *Lactobacillaceae* and *Bifidobacteriaceae*, while inhibiting pathogens like *Clostridiaceae* and *Escherichia coli* [148,241]. Both probiotics and prebiotics exhibit anti-inflammatory and immunomodulatory properties [148,241,242], improving gut barrier function [239,243]. The majority of published studies suggest that supplementation of these compounds may counteract age-related muscle decline by increasing the production of *Bifidobacteriaceae* and butyrate in older people [244,245]. However, studies focused specifically on frail or sarcopenic subjects are still lacking and the majority of evidence came from animal studies.

Other dietary compounds like omega-3 polyunsaturated fatty acids (PUFAs) are gaining growing interest in both the oral and gut microbiota modulation, with implications on frailty and sarcopenia underlying mechanisms. PUFAs, particularly through their derivatives known as specialized pro-resolving mediators (SPMs), help resolve inflammation and reduce ROS production at the mitochondrial level [246], maintaining tissue homeostasis in the oral cavity [247] and in the gut [248], also decreasing muscle protein breakdown in the skeletal muscle [246]. The SPMs are generated from dietary omega-3 PUFAs intake and are found in various body fluids and tissues, including saliva, gingiva, and gingival crevicular fluid [247]. By interacting with immune cells, SPMs can modulate inflammatory responses in periodontal tissues [247]. This immune modulation is linked to changes in the composition of oral bacteria, potentially shifting the microbial community toward a healthier balance. Some in vitro studies also suggest that omega-3 PUFAs have direct antibacterial properties, further influencing the oral microbiota by inhibiting the proliferation of various oral bacteria, including *S. mutans*, *C. albicans*, *A. actinomycetemcomitans*, *F. nucleatum*, and *P. gingivalis* [249,250]. Stańdo-Retecka et al. [251] in a randomized controlled trial, found that supplementation with a high dose of fish oil containing omega-3 PUFAs during non-surgical treatment in stage III or IV periodontitis patients was associated with a reduced number of periodontal pathogens like *P. gingivalis*, *T. forsythia*, *Treponema denticola* and *Aggregatibacter actinomycetemcomitans.* Understanding how SPMs interact with the oral microbiota may lead to new biomarkers for periodontal inflammation and innovative microbiota-targeted therapies [247].

Omega-3 PUFAs play a significant role in modulating gut microbiota composition and related health outcomes too. Omega-3 PUFAs have been suggested to restore microbial balance in certain pathological conditions including cardiovascular disease by increasing the production of SCFAs and modulation of inflammation [248]. Omega-3 PUFAs also exert an influence on the abundance of dominant gut bacterial phyla including Bacillota and Bacteroidota by maintaining a healthy Bacillota/Bacteroidota ratio, a key marker of gut health [252]. However, an imbalanced intake of omega-3 versus omega-6 PUFAs can disrupt this ratio, potentially contributing to conditions such as obesity and non-alcoholic fatty liver disease [252].

However, in some long-term studies, omega-3 supplementation and dietary interventions did not significantly alter markers of gut-related inflammation or cardiovascular outcomes, suggesting that individual responses may vary depending on the dose and period of supplementation as well as disease state [246,248].

### 5.2. Exercise Strategies

Some evidence, mainly coming from animal studies, supports that physical exercise may influence microbiota composition and modulate inflammation [143,253]. On the other hand, only a few studies (especially observational) have shown a gut modification after exercise in humans [143]. Moderate exercise has been reported to increase intestinal motility, which is known to affect gut microbiota in humans [254,255,256]. Endurance exercise has been indicated as inducing mitochondrial biogenesis, preventing mitochondrial DNA depletion and mutations, and increasing mitochondrial oxidative and antioxidant capacity [142]. However, stressful conditions such as overtraining can lead to inflammation in the gastrointestinal tract, favoring LPS translocation and the proliferation of pathobionts [142]. Existing studies are limited by heterogeneity in exercise protocols, small sample sizes, and short intervention durations, making it difficult to establish definitive causal relationships and to identify optimal exercise regimens for gut microbiota modulation.

Beneficial effects of physical exercise on oral microbiota composition in older people have also been reported. Lavilla-Lerma et al. [257], in a randomized controlled trial involving older people, investigated the effects of moderate-intensity continuous training and high-intensity interval training in the modulation of oral microbiota. The authors found that high-intensity interval training resulted in significant temporal changes in the Richness index, as well as in a notable decrease in Simpson and Shannon diversity indices [257]. Conversely, moderate-intensity continuous training was associated with an increasing trend in Simpson and Shannon indices over time, along with a reduction in Bacillota and an elevation in Bacteroidota levels [257]. Additionally, significant alterations in the abundance of pathogenic species were observed following completion of either exercise intervention [257]. However, most evidence came from studies conducted in young adult athletes [258], with few studies specifically focused on older adults’ oral microbiota. Current research on the effects of physical exercise on oral microbiota is limited by small sample sizes and predominantly observational designs. Future studies should thus focus on longitudinal, controlled trials to elucidate causal mechanisms and explore the impact of different exercise modalities on oral microbial health.

## 6. Research Limitations and Future Perspectives

### 6.1. Research Limitations

Research on the oral–gut microbiota axis is limited by the complexity of microbial translocation pathways, challenges in distinguishing transient from colonizing microbes, and a lack of standardized methodologies for sampling and sequencing across different sites, making it difficult to define causal interactions and their impact on systemic health. In the context of aging and disease, studying the oral–gut microbiota axis is limited by age-related changes in immunity, microbiota composition, and mucosal barrier integrity, which complicate the interpretation of microbial translocation and bidirectional interactions. Additionally, the presence of concomitant diseases and the use of multiple medications in older populations further challenge the ability to establish causal links between microbial shifts and disease progression.

Research on the oral–gut microbiota axis is also constrained by observational and cross-sectional study designs, as well as by small human sample sizes and reliance on animal models that may not fully replicate human aging or disease processes, further limiting the generalizability and interpretation of findings. The lack of specific studies on the association between salivary microbiota and sarcopenia also represents a significant knowledge gap.

Most published studies on the efficacy of probiotics and prebiotics in counteracting age-related muscle decline and frailty come from animal studies, with very limited studies specifically focused on frail or sarcopenic subjects. Despite growing evidence supporting the benefits of probiotics for oral health, additional clinical research is needed to establish their long-term efficacy and optimal treatment protocols. The relationship between protein intake with oral gut microbiota composition is very complex and not yet fully understood. It is probably influenced by factors such as protein quality and microbiota metabotype, with microbiota-mediated anabolic responses to protein intake/supplementation probably differing on an individual basis.

### 6.2. Future Perspectives

The oral–gut microbiota axis is an emerging field of research. However, there are several research gaps limiting our current understanding. Oral and gut microbiota are frequently studied in isolation. By examining their combined interactions, including signaling pathways, molecular mechanisms, as well as microbial metabolites involved, could provide a holistic approach in understanding the oral and gut microbiota impact on human health. Multi-omics investigations of the salivary and gut microbiota could bridge the knowledge gap on their associations with frailty and sarcopenia. A better knowledge of the oral–gut microbiota axis could also offer promising insights for nutritional interventions and therapeutic strategies for age-related muscle decline, frailty, and maintenance of systemic health. The recognition that microbiota-mediated anabolic responses to protein intake/supplementation may differ on an individual basis suggests the need for more personalized strategies based on microbiota profiles.

## 7. Conclusions

The interaction between the oral and gut microbiota is a complex and evolving area of research that underscores the interdependence of microbial communities across various body sites. Emerging evidence suggests that the oral and gut microbiota influence each other through several mechanisms, including the translocation of microorganisms via saliva, the bloodstream and the fecal-oral route. Disruptions in oral and/or gut microbiota, such as those caused by oral diseases, antibiotics, or gastrointestinal conditions, can lead to dysbiosis, which in turn may contribute to increased systemic inflammation and mitochondrial dysfunction promoting the development of frailty and accelerated muscle decline. Among various lifestyle factors, diet plays a significant role in shaping both oral and gut microbiota, with a significant contribution to inflammatory status and mitochondrial health. Healthy dietary patterns constituted by low consumption of ultra-processed foods (rich in refined sugars and saturated and trans fatty acids), limited alcohol intake, as well as supplementation with prebiotics, probiotics and omega-3 PUFAs seem to constitute common strategies promoting both oral and gut microbiota health, with potential beneficial effects also on frailty and sarcopenia. Furthermore, adequate protein intake seems to promote the homeostasis of both microbiota, thus suggesting microbiota-driven beneficial effects on muscle parameters and frailty. Further research exploring oral–gut interactions and their modulation may pave the way to the development of novel therapeutic strategies based on nutritional approaches aimed at counteracting both frailty and sarcopenia.

## Figures and Tables

**Figure 1 nutrients-17-02408-f001:**
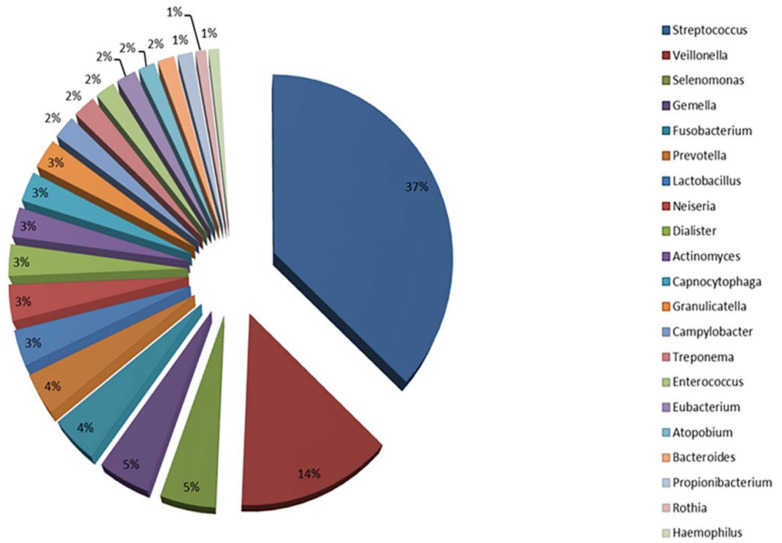
The main relative members of the phyla of oral microbiota. Modified from Santacroce et al. [27] under the terms of the Creative Commons Attribution—NonCommercial 4.0 License https://creativecommons.org/licenses/by-nc/4.0/ (accessed on 5 May 2025) which permits non-commercial use, reproduction and distribution of the work without further permission provided the original work is attributed. Source: HOMD, http://www.homd.org/ (accessed on 5 May 2025).

**Figure 2 nutrients-17-02408-f002:**
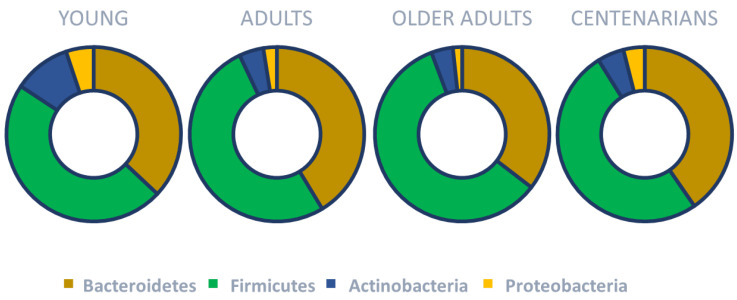
Main gut microbiota changes across the lifespan. Based on the concepts and findings of Haran and McCormick [66], Biagi et. al. [67] and Monira et al. [68].

**Figure 3 nutrients-17-02408-f003:**
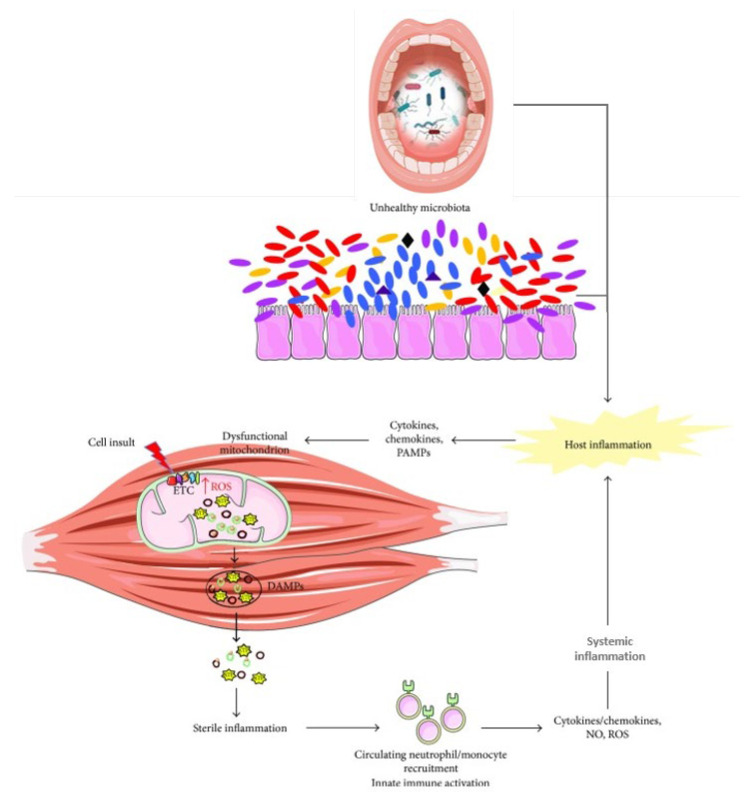
Relationship between mitochondrial dysfunction and inflammation in muscle wasting. Modified from Picca et al. [1] under the Creative Commons Attribution License, which permits unrestricted use, distribution, and reproduction in any medium, provided the original work is properly cited.

**Table 1 nutrients-17-02408-t001:** Oral and gut microbiota changes with aging.

Oral Microbiota	Gut Microbiota
**Older people**	
**↑** *Lactobacillaceae*, *Streptococcus anginosus*, and *Gemella sanguinis* **↓** *Neisseria*	**↓** Bacillota/Bacteroidota ratio, *Bifidobacteriaceae* **↑** Pseudomonadota (*Escherichia coli*, *Klebsiella*, *Acquabacterium*)
***Denture users:* ↑** Bacillota and Actinomycetota	-
***Edentulous:* ↑** *Prevotella histicola*, *Veillonella atypica*, *Streptococcus salivarius*, and *Streptococcus parasanguinis*	-
**Centenarians**	
**Toothy centenarians** ***Dental plaque and saliva:* ↑** Spirochaetota and Synergistota (at phylum level), *Aggregatibacter* spp., *Prevotella* spp., *Campylobacter* spp., *Anaeroglobus* spp., *Selenomonas* spp., *Fusobacterium* spp., and *Porphyromonas endodontalis* (at genus level) ***Dental plaque:* ↑** *Bifidobacterium* and *Scardovia* (at genus level), *Porphyromonas gingivalis*, *Tannerella forsythia*, and *Prevotella intermedia* (at species level) **Edentulous** ***Dental plaque and saliva:* ↑** Bacillota and Actinomycetota (at phylum level), *Streptococcus* spp. (at genus level)	**↑** Pseudomonadota (*Escherichia coli* et rel., *Haemophilus* spp., *Klebsiella pneumoniae* et rel., *Leminorella* spp., *Proteus* et rel., *Pseudomonas*, *Serratia* spp., *Vibrio* spp., *and Yersinia* et rel.), Bacillota (*Bacillus* spp., *Staphylococcus* spp.) **↑** *Methanobrevibacter smithii*, *Bifidobacterium adolescentis*, *Clostridium leptum* **↑** Lactic acid species (*Lactobacillaceae*) **↓** Bacillota/>Bacteroidota ratio **↓** *Faecalibacterium prausnitzii*, *Agathobacter rectalis*

**↑** Abundance; **↓** Depletion.

## Data Availability

No new data were created or analyzed in this study. Data sharing is not applicable to this article.

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
