# Peer review of "The Oral–Gut Microbiota Axis as a Mediator of Frailty and Sarcopenia"

_nutrients, 2025, doi:10.3390/nu17152408_

Round 1
Reviewer 1 Report
Comments and Suggestions for Authors
Journal
Nutrients (ISSN 2072-6643)
Manuscript ID
nutrients-3745031
Type
Review
Title
The oral-gut microbiota axis as a mediator of frailty and sarcopenia.
Section: Geriatric Nutrition
Special Issue: Addressing Malnutrition in the Aging Population
___________________________________
OVERALL COMMENTS
Based on the statement that gut microbiota is traditionally studied in isolation, the oral microbiota has been recognized as interconnected through anatomical and physiological pathways. Both undergo changes with aging, characterized by a decline in microbial diversity and a shift toward potentially harmful species. Moreover, they say that alterations in the oral-gut microbiota axis contribute to inflamm-ageing and mitochondrial dysfunction, which are key mechanisms underlying frailty and sarcopenia. Metabolites, such as short-chain fatty acids and modified bile acids, appear to play an emerging role in influencing microbial homeostasis and muscle metabolism. In this sense, the authors intended to investigate the relationship between frailty and sarcopenia.
TITLE
The title is fine.
ABSTRACT
The Abstract lacks the aims of the study and the results of the search. Conclusions are always informative. I suggest including. In lines 84-85 we can read “The aim of this review is therefore to provide an overview of oral-gut communications in mediating frailty and sarcopenia.” The authors could use this sentence to show the aim of the study.
_______
KEYWORDS
The authors presented the following keywords:
Nutrition; Diet; Microbiome; Malnutrition; Aging; Skeletal Muscle; Inflammation; Mitochondria
I suggest: nutrition; diet; oral microbiota; gut microbioma; aging; frailty; Inflammation; Mitochondrial dysfunction
INTRODUCTION AND THE ENTIRE TEXT
I would like to see more references published in 2025 included.
Secondly, as the oral microbiota is less studied than the gut microbiota, I think the authors should provide a more detailed explanation of the oral microbiota in sarcopenic processes.
In topic 2.2, the authors explore changes in the microbiota with advancing age. What would be the predominant factors for these changes?
I suggest inserting a table that compares the microorganisms of the oral and gut microbiota. This would make it easier for the reader to understand.
I suggest correlating the link to oral-gut dysbiosis with greater depth.
I also suggest that the authors do a critical analysis of the studies. For example, some have a very small sample size, which limits the findings.
In lines 228- 234 we can read that: “In IBD, bile acid malabsorption and receptor alterations may facilitate the migration of oral pathobionts to the gut. Proposed mechanisms by which oral bacteria contribute to IBD include [94]: 231 1. Disruption of intestinal barriers (P. gingivalis, K. pneumoniae), 2. Lipopolysaccharide-triggered inflammation (F. nucleatum, K. pneumoniae), 3. T cell imbalances (F. nucleatum, Candida albicans), 4. Inflammasome activation and immune dysregulation.”
I suggest that the authors briefly expand on each item to provide the reader with a better understanding of why these factors may interfere with inflammatory bowel disease.
- nucleatum, Candida albicans should be in italics. Please check italics in the entire manuscript.
I suggest including a table summarizing the findings from clinical trials and another showing the findings from animal studies. This would avoid confusion when reading the text.
In lines 315-317 we can find that: Ogawa et al. found a higher relative abundance of Actinomyces, Streptococcus, Bacilli, Selenomonas, Veillonella, and Haemophilus taxa, and a decreased relative abundance of Prevotella, Leptotrichia, Campylobacter, and Fusobacterium. Another study from Wells et al., reported a positive association between frailty and decreased salivary microbiota diversity. However, to our knowledge, no study has explored associations between salivary microbiota composition and sarcopenia.
Is the reference for Wells 136? And for Ogawa et al.?
The citations in brackets: the font size seems to be bigger than the text.
The section: 5. Dietary and exercise strategies targeting the oral and gut microbiota and their effects on frailty and sarcopenia is long.
I suggest splitting into one that describes the role of the exercise and another just about nutrition.
FINAL SUGGESTIONS:
I suggest including a separate paragraph showing the limitations for this study, and another for Future perspectives:
How can this review contribute to further research?
What comes next?
What kind of studies do the authors suggest to corroborate (or not) their results?
____________
REFERENCES
As mentioned earlier, please include more recent references.
Author Response
Responses to Reviewer 1
OVERALL COMMENTS
Based on the statement that gut microbiota is traditionally studied in isolation, the oral microbiota has been recognized as interconnected through anatomical and physiological pathways. Both undergo changes with aging, characterized by a decline in microbial diversity and a shift toward potentially harmful species. Moreover, they say that alterations in the oral-gut microbiota axis contribute to inflamm-ageing and mitochondrial dysfunction, which are key mechanisms underlying frailty and sarcopenia. Metabolites, such as short-chain fatty acids and modified bile acids, appear to play an emerging role in influencing microbial homeostasis and muscle metabolism. In this sense, the authors intended to investigate the relationship between frailty and sarcopenia.
Re: Thank you very much for reviewing of our manuscript. Your comments and suggestions certainly add value to our paper.
TITLE
The title is fine.
ABSTRACT
The Abstract lacks the aims of the study and the results of the search. Conclusions are always informative. I suggest including. In lines 84-85 we can read “The aim of this review is therefore to provide an overview of oral-gut communications in mediating frailty and sarcopenia.” The authors could use this sentence to show the aim of the study.
Re: Thank you. We included the following statement in the abstract section (Now line 39-42): The aim of this review is therefore to provide an overview of oral-gut communications in mediating frailty and sarcopenia. PubMed, EMBASE and Scopus databases were searched for relevant articles. We limited our search to manuscripts published in the English language.
_______
KEYWORDS
The authors presented the following keywords:
Nutrition; Diet; Microbiome; Malnutrition; Aging; Skeletal Muscle; Inflammation; Mitochondria
I suggest: nutrition; diet; oral microbiota; gut microbioma; aging; frailty; Inflammation; Mitochondrial dysfunction
Re: Thank you. We revised the keywords accordingly.
INTRODUCTION AND THE ENTIRE TEXT
I would like to see more references published in 2025 included.
Re: Thank you. We added some references published in 2025 as well as those published in the past 5 years.
2025: doi.org/10.1016/j.gene.2025.149433; doi.org/10.3390/microorganisms13040814; doi.org/10.1186/s10020-025-01166-w; doi.org/10.1038/s41522-025-00646-5; doi.org/10.7759/cureus.78168; doi.org/10.1002/jcsm.13750; doi.org/10.1016/j.jnha.2025.100582
doi.org/10.1016/j.aggp.2025.100142
2024: doi.org/10.3389/fcvm.2024.1406220; doi.org/10.1080/19490976.2024.2333463; doi.org/10.1093/ecco-jcc/jjae028; doi.org/10.1038/s41430-024-01486-w; doi.org/10.3390/ijms252312815; doi.org/10.1016/j.healun.2024.04.069; doi.org/10.1016/j.maturitas.2024.107973
Secondly, as the oral microbiota is less studied than the gut microbiota, I think the authors should provide a more detailed explanation of the oral microbiota in sarcopenic processes.
Re: Thank you. We added more information about the role of oral microbiota in the sarcopenic process in section 4. Influence of the oral-gut axis on frailty and sarcopenia (Lines 329-356, lines 368-370, lines 397-438). These include mechanisms and signaling pathways, also according to Reviewer 2 suggestions, studies that explored associations between both oral microbiota alterations with frailty and sarcopenia, as well as associations between certain dental problems frequently characterized by oral dysbiosis with frailty and sarcopenia.
In topic 2.2, the authors explore changes in the microbiota with advancing age. What would be the predominant factors for these changes?
I suggest inserting a table that compares the microorganisms of the oral and gut microbiota. This would make it easier for the reader to understand.
Re: Thank you. We added a table (now Table 1) comparing the oral and gut microbiota composition with advancing age, also according to Reviewer 2 comment. Furthermore, we revised the part of the topic 2.2 highlighting the predominant factors for age related changes in gut microbiota as follows “Age-related microbial shifts are therefore influenced not only by chronological age but also by several factors such as the use of multiple medications including antibiotics, proton pump inhibitors, and laxatives use, chronic conditions, immune system aging (i.e., immunosenescence), increased gut permeability to lipopolysaccharides, and changes in diet and lifestyle, but also by individual genetic predisposition”.
I suggest correlating the link to oral-gut dysbiosis with greater depth.
Re: We added some additional information about the mechanisms underlying oral-gut dysbiosis. These include signaling pathways, microbe-host interactions, and key immune mediators like cytokines and transcription factors. This was also done according to Reviewer 2 suggestions. We added this part in section 4. Influence of the oral-gut axis on frailty and sarcopenia.
I also suggest that the authors do a critical analysis of the studies. For example, some have a very small sample size, which limits the findings.
Re: Done. We addressed this point by adding limitations due to small sample sizes in the paragraph about limitations, as you suggested. We also added findings of some studies characterized by bigger sample sizes.
In lines 228- 234 we can read that: “In IBD, bile acid malabsorption and receptor alterations may facilitate the migration of oral pathobionts to the gut. Proposed mechanisms by which oral bacteria contribute to IBD include [94]: 231 1. Disruption of intestinal barriers (P. gingivalis, K. pneumoniae), 2. Lipopolysaccharide-triggered inflammation (F. nucleatum, K. pneumoniae), 3. T cell imbalances (F. nucleatum, Candida albicans), 4. Inflammasome activation and immune dysregulation.”
I suggest that the authors briefly expand on each item to provide the reader with a better understanding of why these factors may interfere with inflammatory bowel disease.
RE: We expanded each item by discussing underlying mechanisms for each point, also according to reviewer 2 suggestions (e.g., signaling pathways, microbe-host interactions, and key immune mediators like cytokines and transcription factors). We moved this paragraph to section 4. Influence of the oral-gut axis on frailty and sarcopenia, highlighting how these mechanisms could influence frailty and sarcopenia.
nucleatum, Candida albicans should be in italics. Please check italics in the entire manuscript.
Re: Thank you. Done.
I suggest including a table summarizing the findings from clinical trials and another showing the findings from animal studies. This would avoid confusion when reading the text.
RE: Thank you. We suppose that you referred to paragraph 4. Influence of the oral-gut axis on frailty and sarcopenia. We highlighted findings coming from animal vs human studies in the text by better specifying if studies were conducted in animal models or human populations. Additionally, as you suggested, we added in the section about limitations that most evidence came from animal studies and small human sample sizes.
In lines 315-317 we can find that: Ogawa et al. found a higher relative abundance of Actinomyces, Streptococcus, Bacilli, Selenomonas, Veillonella, and Haemophilus taxa, and a decreased relative abundance of Prevotella, Leptotrichia, Campylobacter, and Fusobacterium. Another study from Wells et al., reported a positive association between frailty and decreased salivary microbiota diversity. However, to our knowledge, no study has explored associations between salivary microbiota composition and sarcopenia.
Is the reference for Wells 136? And for Ogawa et al.?
RE: Done. We added also in the line indicated the reference for Ogawa et al. (now ref. N° 166)
The citations in brackets: the font size seems to be bigger than the text.
Re: Thank you. Probably the reference manager tool we used made the font size bigger than the text. We revised it.
The section: 5. Dietary and exercise strategies targeting the oral and gut microbiota and their effects on frailty and sarcopenia is long.
I suggest splitting into one that describes the role of the exercise and another just about nutrition.
Re: Thank you. We divided section 5 into 5.1 Dietary strategies and 5.2 Exercise strategies. We also added in the section about exercise strategies the findings of a recent randomized controlled trial (published in 2024) exploring the effects of exercise on oral microbiota composition in older adults.
FINAL SUGGESTIONS:
I suggest including a separate paragraph showing the limitations for this study, and another for Future perspectives:
How can this review contribute to further research?
What comes next?
What kind of studies do the authors suggest to corroborate (or not) their results?
Re: Thank you. We added a paragraph about the limitations of this review (6. Research limitations and future perspectives, split into 6.1. Research limitations and 6.2. Future perspectives).
REFERENCES
As mentioned earlier, please include more recent references.
RE: We added some references published in 2025 and in the last 5 years.
2025: doi.org/10.1016/j.gene.2025.149433; doi.org/10.3390/microorganisms13040814; doi.org/10.1186/s10020-025-01166-w; doi.org/10.1038/s41522-025-00646-5; doi.org/10.7759/cureus.78168; doi.org/10.1002/jcsm.13750; doi.org/10.1016/j.jnha.2025.100582
doi.org/10.1016/j.aggp.2025.100142
2024: doi.org/10.3389/fcvm.2024.1406220; doi.org/10.1080/19490976.2024.2333463; doi.org/10.1093/ecco-jcc/jjae028; doi.org/10.1038/s41430-024-01486-w; doi.org/10.3390/ijms252312815; doi.org/10.1016/j.healun.2024.04.069; doi.org/10.1016/j.maturitas.2024.107973
Reviewer 2 Report
Comments and Suggestions for Authors
nutrients-3745031
Type: Review
Title: The oral-gut microbiota axis as a mediator of frailty and sarcopenia.
Authors: Domenico Azzolino *, Margherita Carnevale Schianca, Lucrezia Bottalico, Marica Colella, Alessia Felicetti, Simone Perna *, Leonardo Terranova, Franklin Garcia Godoy, Mariangela Rondanelli, Pier Carmine Passarelli *, Tiziano Lucchi
This review explores the emerging concept of the oral-gut microbiota axis and its implications for age-related diseases, particularly frailty and sarcopenia. Given the growing burden of global population aging and age-related musculoskeletal decline, this review is timely, conceptually novel, and highly relevant.
While the gut microbiota has been widely studied in relation to frailty and sarcopenia, our review provides a novel and under-researched perspective by incorporating the oral microbiota into this paradigm and highlighting their interconnectedness through multiple physiological pathways. This conceptual framework linking oral health, microbial dysbiosis, inflammation, and sarcopenia is expected to make a valuable contribution to gerontology and microbiome research. However, several issues need further revision as noted below.
[Major concerns]
- In the case of sarcopenia, since WHO first assigned a disease code in 2016, many countries have assigned their own codes. However, there is still no appropriate treatment. As a researcher studying sarcopenia, I am curious about whether Italy has assigned a disease code, sarcopenia epidemiological data, and the current status of status of drug development in Italy and in the world. Therefore, I think it would be appropriate to describe these contents first in an appropriate place in the beginning of the paper.
- Lack of mechanistic depth: While this review highlights pathways such as inflammation and mitochondrial dysfunction, the mechanistic basis is very superficial. Including more molecular-level details (e.g., signaling pathways, microbe-host interactions, involvement of specific cytokines or transcription factors) would likely enhance the scientific depth.
- Oral microbiota changes with aging: Figure 2 summarizes the changes in intestinal bacteria with aging, but it would be a good idea to add a figure or table related to oral microbiota changes with aging in Section 2.1.
- English: The English writing of the paper is not problematic overall, but there are some words that are unnecessarily capitalized or in italics but are not. I hope you will do a proper proofreading once again. Examples: Tumor Necrosis Factor alfa (TNF-α) and Interleukin-1 beta (IL-1β) at Line 292; etc.
- Abbreviations: Abbreviations can enhance clarity and conciseness but should be used only for terms repeated frequently. Define each abbreviation by writing the full term followed by the abbreviation in parentheses at first mention, then use the abbreviation consistently throughout the paper. Define abbreviations separately in the Abstract and main text, as the Abstract is often read on its own. Use abbreviations in the Abstract only if they appear more than once. Proofread carefully to avoid redundant or inconsistent abbreviation use.
- In cases where abbreviations are used within figures or tables, please list these abbreviations along with their corresponding full names in the figure legends or at the bottom of corresponding tables. If there are two or more abbreviations, arrange them in alphabetical order. In this case, non-proper nouns should not have their first letters capitalized. And when listing abbreviations and full names, rewrite them according to the following examples at Figure 3. DAMPs, damage-associated molecular patterns; ETC, electron transport chain; NO, nitric oxide; PAMPs, pathogen-associated molecular patterns; ROS, reactive oxygen species.
[Minor concerns]
- Line 113: Rewrite ‘Human Papillomavirus’ as ‘human papillomavirus’ or ‘human papilloma virus’.
- Figure 3: ‘sistemic inflammation’ should be written as ‘systemic inflammation’.
- Line 228: Define HIV.
- Line 234: ‘ nucleatum and Candida albicans’ should be written in italics.
- Line 292: Rewrite ‘Tumor Necrosis Factor alfa (TNF-α) and Interleukin-1 beta (IL-1β)’.
- Line 389: Define GI.
- Line 411: Define ASV.
Overall, the manuscript can be considered to publication after major revision as indicated above.

nutrients-3745031
Type: Review
Title: The oral-gut microbiota axis as a mediator of frailty and sarcopenia.
Authors: Domenico Azzolino *, Margherita Carnevale Schianca, Lucrezia Bottalico, Marica Colella, Alessia Felicetti, Simone Perna *, Leonardo Terranova, Franklin Garcia Godoy, Mariangela Rondanelli, Pier Carmine Passarelli *, Tiziano Lucchi
This review explores the emerging concept of the oral-gut microbiota axis and its implications for age-related diseases, particularly frailty and sarcopenia. Given the growing burden of global population aging and age-related musculoskeletal decline, this review is timely, conceptually novel, and highly relevant.
While the gut microbiota has been widely studied in relation to frailty and sarcopenia, our review provides a novel and under-researched perspective by incorporating the oral microbiota into this paradigm and highlighting their interconnectedness through multiple physiological pathways. This conceptual framework linking oral health, microbial dysbiosis, inflammation, and sarcopenia is expected to make a valuable contribution to gerontology and microbiome research. However, several issues need further revision as noted below.
[Major concerns]
- In the case of sarcopenia, since WHO first assigned a disease code in 2016, many countries have assigned their own codes. However, there is still no appropriate treatment. As a researcher studying sarcopenia, I am curious about whether Italy has assigned a disease code, sarcopenia epidemiological data, and the current status of status of drug development in Italy and in the world. Therefore, I think it would be appropriate to describe these contents first in an appropriate place in the beginning of the paper.
- Lack of mechanistic depth: While this review highlights pathways such as inflammation and mitochondrial dysfunction, the mechanistic basis is very superficial. Including more molecular-level details (e.g., signaling pathways, microbe-host interactions, involvement of specific cytokines or transcription factors) would likely enhance the scientific depth.
- Oral microbiota changes with aging: Figure 2 summarizes the changes in intestinal bacteria with aging, but it would be a good idea to add a figure or table related to oral microbiota changes with aging in Section 2.1.
- English: The English writing of the paper is not problematic overall, but there are some words that are unnecessarily capitalized or in italics but are not. I hope you will do a proper proofreading once again. Examples: Tumor Necrosis Factor alfa (TNF-α) and Interleukin-1 beta (IL-1β) at Line 292; etc.
- Abbreviations: Abbreviations can enhance clarity and conciseness but should be used only for terms repeated frequently. Define each abbreviation by writing the full term followed by the abbreviation in parentheses at first mention, then use the abbreviation consistently throughout the paper. Define abbreviations separately in the Abstract and main text, as the Abstract is often read on its own. Use abbreviations in the Abstract only if they appear more than once. Proofread carefully to avoid redundant or inconsistent abbreviation use.
- In cases where abbreviations are used within figures or tables, please list these abbreviations along with their corresponding full names in the figure legends or at the bottom of corresponding tables. If there are two or more abbreviations, arrange them in alphabetical order. In this case, non-proper nouns should not have their first letters capitalized. And when listing abbreviations and full names, rewrite them according to the following examples at Figure 3. DAMPs, damage-associated molecular patterns; ETC, electron transport chain; NO, nitric oxide; PAMPs, pathogen-associated molecular patterns; ROS, reactive oxygen species.
[Minor concerns]
- Line 113: Rewrite ‘Human Papillomavirus’ as ‘human papillomavirus’ or ‘human papilloma virus’.
- Figure 3: ‘sistemic inflammation’ should be written as ‘systemic inflammation’.
- Line 228: Define HIV.
- Line 234: ‘ nucleatum and Candida albicans’ should be written in italics.
- Line 292: Rewrite ‘Tumor Necrosis Factor alfa (TNF-α) and Interleukin-1 beta (IL-1β)’.
- Line 389: Define GI.
- Line 411: Define ASV.
Overall, the manuscript can be considered to publication after major revision as indicated above.
Author Response
Responses to Reviewer 2
This review explores the emerging concept of the oral-gut microbiota axis and its implications for age-related diseases, particularly frailty and sarcopenia. Given the growing burden of global population aging and age-related musculoskeletal decline, this review is timely, conceptually novel, and highly relevant.
While the gut microbiota has been widely studied in relation to frailty and sarcopenia, our review provides a novel and under-researched perspective by incorporating the oral microbiota into this paradigm and highlighting their interconnectedness through multiple physiological pathways. This conceptual framework linking oral health, microbial dysbiosis, inflammation, and sarcopenia is expected to make a valuable contribution to gerontology and microbiome research. However, several issues need further revision as noted below.
Re: Thank you for the time spent reviewing our manuscript and for your valuable suggestions to improve our manuscript, especially in the context of sarcopenia. We hope that our revision, based on your feedback, has improved the quality of the work.
[Major concerns]
In the case of sarcopenia, since WHO first assigned a disease code in 2016, many countries have assigned their own codes. However, there is still no appropriate treatment. As a researcher studying sarcopenia, I am curious about whether Italy has assigned a disease code, sarcopenia epidemiological data, and the current status of status of drug development in Italy and in the world. Therefore, I think it would be appropriate to describe these contents first in an appropriate place in the beginning of the paper.
Re: Thank you very much for this interesting point. We agree with you about mentioning the ICD-10 code for sarcopenia. According to the International Classification of Diseases, Tenth Revision (ICD-10), sarcopenia is identified by the code M62.84, which is adopted by over 100 countries, including Italy, that utilize the ICD-10 system. However, in some Italian regions, including the Lombardy region, the ICD-9 is still widely used in clinical practice. Indeed, it has been suggested that sarcopenia could be assigned the code 728.2 (muscle wasting and atrophy, not elsewhere classified, unspecified site) according to the ICD-9. We highlighted this point in the introduction. We also briefly discussed the status of drug development for sarcopenia, and, regarding epidemiological data, we added some data about the prevalence of sarcopenia (“The prevalence of sarcopenia varies widely between studies and depending on the operational definition used. We mentioned a recent systematic review of meta-analyses that estimated a worldwide prevalence of sarcopenia of about 10-16 % in older people” DOI: 10.1016/j.metabol.2023.155533).
Lack of mechanistic depth: While this review highlights pathways such as inflammation and mitochondrial dysfunction, the mechanistic basis is very superficial. Including more molecular-level details (e.g., signaling pathways, microbe-host interactions, involvement of specific cytokines or transcription factors) would likely enhance the scientific depth.
RE: Thank you for this important point. We expanded the discussion about the proposed pathways also according to Reviewer 1 suggestions in paragraph 4. Influence of the oral-gut axis on frailty and sarcopenia. In particular, we added signalic pathways, microbe-host interactions, as well as specific cytokines or transcription factors involved in the oral-gut communication axis and in the context of frailty and sarcopenia.
Oral microbiota changes with aging: Figure 2 summarizes the changes in intestinal bacteria with aging, but it would be a good idea to add a figure or table related to oral microbiota changes with aging in Section 2.1.
Re: We added a table comparing oral and gut microbiota changes with aging, also according to Reviewer 1 suggestions.
English: The English writing of the paper is not problematic overall, but there are some words that are unnecessarily capitalized or in italics but are not. I hope you will do a proper proofreading once again. Examples: Tumor Necrosis Factor alfa (TNF-α) and Interleukin-1 beta (IL-1β) at Line 292; etc.
Re: Thank you. We revised it in the entire paper.
Abbreviations: Abbreviations can enhance clarity and conciseness but should be used only for terms repeated frequently. Define each abbreviation by writing the full term followed by the abbreviation in parentheses at first mention, then use the abbreviation consistently throughout the paper. Define abbreviations separately in the Abstract and main text, as the Abstract is often read on its own. Use abbreviations in the Abstract only if they appear more than once. Proofread carefully to avoid redundant or inconsistent abbreviation use.
In cases where abbreviations are used within figures or tables, please list these abbreviations along with their corresponding full names in the figure legends or at the bottom of corresponding tables. If there are two or more abbreviations, arrange them in alphabetical order. In this case, non-proper nouns should not have their first letters capitalized. And when listing abbreviations and full names, rewrite them according to the following examples at Figure 3. DAMPs, damage-associated molecular patterns; ETC, electron transport chain; NO, nitric oxide; PAMPs, pathogen-associated molecular patterns; ROS, reactive oxygen species.
Re: Thank you. We carefully revised the paper for abbreviations.
[Minor concerns]
Line 113: Rewrite ‘Human Papillomavirus’ as ‘human papillomavirus’ or ‘human papilloma virus’.
Re: Thank you. Done.
Figure 3: ‘sistemic inflammation’ should be written as ‘systemic inflammation’.
Re: Thank you. Done.
Line 228: Define HIV.
Re: Thank you. Done.
Line 234: ‘ nucleatum and Candida albicans’ should be written in italics.
Re: Thank you. Done.
Line 292: Rewrite ‘Tumor Necrosis Factor alfa (TNF-α) and Interleukin-1 beta (IL-1β)’.
Re: Thank you. Done.
Line 389: Define GI.
Re: Thank you. Done.
Line 411: Define ASV.
Re: Thank you. We defined ASV at line 247 for the first time.
Overall, the manuscript can be considered to publication after major revision as indicated above.
Re: Thank you very much for reviewing our manuscript.
Round 2
Reviewer 2 Report
Comments and Suggestions for Authors
nutrients-3745031-v2
Type: Review
Title: The oral-gut microbiota axis as a mediator of frailty and sarcopenia.
Authors: Domenico Azzolino *, Margherita Carnevale Schianca, Lucrezia Bottalico, Marica Colella, Alessia Felicetti, Simone Perna *, Leonardo Terranova, Franklin Garcia Godoy, Mariangela Rondanelli, Pier Carmine Passarelli *, Tiziano Lucchi
Most of the points I pointed out in the last review have been corrected appropriately, and the quality of the paper has improved greatly. However, there are still a few minor typos. For example: E. Faecalis should be written as E. faecalis at Lines 113 and 116; etc.
These should be corrected during the proofreading process.
Overall, accept in present form after minor correction.
Comments on the Quality of English Languagenutrients-3745031-v2
Type: Review
Title: The oral-gut microbiota axis as a mediator of frailty and sarcopenia.
Authors: Domenico Azzolino *, Margherita Carnevale Schianca, Lucrezia Bottalico, Marica Colella, Alessia Felicetti, Simone Perna *, Leonardo Terranova, Franklin Garcia Godoy, Mariangela Rondanelli, Pier Carmine Passarelli *, Tiziano Lucchi
Most of the points I pointed out in the last review have been corrected appropriately, and the quality of the paper has improved greatly. However, there are still a few minor typos. For example: E. Faecalis should be written as E. faecalis at Lines 113 and 116; etc.
These should be corrected during the proofreading process.
Overall, accept in present form after minor correction.
Author Response
Response to reviewer 2
Most of the points I pointed out in the last review have been corrected appropriately, and the quality of the paper has improved greatly. However, there are still a few minor typos. For example: E. Faecalis should be written as E. faecalis at Lines 113 and 116; etc.
These should be corrected during the proofreading process.
Re: Thank you very much. We revised the paper accordingly.
Overall, accept in present form after minor correction.